# Effectiveness of the Rehabilitation Training Combined with Maitland Mobilization for the Treatment of Chronic Ankle Instability: A Randomized Controlled Trial

**DOI:** 10.3390/ijerph192215328

**Published:** 2022-11-20

**Authors:** Yikun Yin, Zhengze Yu, Jialin Wang, Junzhi Sun

**Affiliations:** 1College of Physical and Health Education, Guangxi Normal University, Guilin 541006, China; 2Institute of Sports Medicine and Health, Chengdu Sport University, Chengdu 610041, China

**Keywords:** Maitland mobilization, chronic ankle instability, balance, joint range of movement, muscle strength

## Abstract

The study aims to determine whether routine rehabilitation training combined with the Maitland mobilization is more effective than routine rehabilitation training alone in patients with chronic ankle instability, intending to provide a novel rehabilitation strategy for chronic ankle instability. A total of 48 subjects were divided into three groups: EG (Maitland mobilization and routine rehabilitation), CG (routine rehabilitation), and SG (sham mobilization and routine rehabilitation). The intervention was performed three times each week for 4 weeks, for a total of 12 sessions. Before and after the intervention, the muscle strength, star excursion balance test (SEBT), weight-bearing dorsiflexion range of motion (WB-DFROM), ankle range of movement, Cumberland ankle instability tool (CAIT), self-comfort visual analog scale (SCS-VAS), and self-induced stability scale (SISS-VAS) were assessed. The results showed that the improvement of SEBT, WB-DFROM, and active ankle range of movement without the pain in EG was more obvious than CG and SG, but the improvement of the self-report of ankle severity and muscle strength was not. Compared with routine rehabilitation training alone, routine rehabilitation training combined with Maitland mobilization for patients with chronic ankle instability may provide more benefit in terms of balance and ankle range of movement than routine rehabilitation alone, but the improvement in muscle strength was not evident enough.

## 1. Introduction

An ankle sprain is one of the most common sports injuries and the recurrence rate of an ankle sprain is the highest in lower limbs sports injuries [1,2]. The incidence of ankle sports injuries ranges from 25% to 50 [3,4]. Additionally, 50% of patients with acute ankle sprains did not seek medical assistance, which led to subsequent sprains being repeated and eventually to chronic ankle instability (CAI) [2]. Typical symptoms of CAI include decreased muscle strength, ligament elasticity, abnormal proprioception, restricted ankle range of movement, etc. A total of 70% of patients, even if they perform routine rehabilitation training at a later stage, would suffer from numerous functional abnormalities [5,6]. CAI severely restricts the exercise ability of the patients, causing traumatic ankle arthritis [7].

Currently, muscle strength training, physical factor treatments, and Kinesio taping combined with external stabilization measures, such as protective gear, taping, and orthoses, are the primary therapeutic approaches for CAI [8,9,10]. Tsikopoulos K et al. [11] found that external stabilization measures did not have more improvements than the controls in dynamic balance through a network meta-analysis. Biz C and Ling W et al. [12,13] found that Kinesio taping did not obviously improve the dynamic balance of CAI patients, especially for ankle proprioception and ankle range of movement, but Kinesio taping can obviously improve the activation of the long peroneal muscle. Luan L et al. [14] explored whether muscle strength training can improve balance performance in patients with CAI. Compared with the controls, muscle strength training did not improve the SEBT results. Hall E.A et al. [15] found that the muscle strength of ankle dorsiflexion, varus, and eversion was obviously improved after strength training for 6 weeks, and the pain was also relieved.

Therefore, routine rehabilitation training can reduce the degree of swelling and pain after the ankle sprain and enhance muscle strength [16], but it cannot address the possible sequelae of ankle sprain, such as proprioception damage and restricted range of movement.

The main characteristic of ankle instability following an ankle sprain, as well as one of the factors contributing to the risk of re-injury, is the restricted range of movement in the ankle, proprioceptive impairment, and decreased neuromuscular control ability [17,18,19,20]. The ankle range of movement is related to the changes in the subtalar joint and the talocrural joint kinematics; that is, the reduction of talus backward sliding or the change of talus position relative to the ankle point [21]. The disorder of ankle proprioceptive and neuromuscular control ability would induce decreased balance ability in CAI patients [22]. Even after routine rehabilitation training, the preceding symptoms may still be existent.

Manual therapy improves and recovers joint physiological and accessory movements through passive activities. Moreover, it can stretch the ligaments and joint capsule around a joint, reduce soft tissue adhesion, realign joints, and recover ligament extensibility [23]. Maitland mobilization was originated by Geoffrey Douglas Maitland, an Australian physiotherapist. It is a crucial diagnostic and therapeutic technique consisting of evaluation and treatment for skeletal muscle system dysfunction. The primary purpose of manipulation, which includes rolling, rotating, sliding, and separation traction, is to improve and recover the physiological movement and auxiliary movement of joints [24,25]. In previous studies on musculoskeletal diseases [25,26,27], manual therapy has been shown to have a positive recovery effect and is regarded as a good way to make the patients comfortable by increasing the joint range of movement, stimulating nerve conduction pathways, and improving proprioception [28,29,30,31].

To reduce the impact of confounding factors and improve the quality of manual therapy research, researchers have previously conducted comparative studies of real manual therapy and sham manual therapy [32]. The objective of the study was as follows: to add Maitland mobilization to routine rehabilitation training for CAI patients to explore whether there was a more positive improvement than routine rehabilitation alone or routine rehabilitation combined with sham manual therapy, and to determine the effectiveness of Maitland mobilization for CAI patients. We hypothesized that the combination of routine rehabilitation training and Maitland mobilization would have a positive effect on balance, ankle range of movement, and muscle strength in patients with CAI, to explore whether Maitland mobilization is synergistic with routine rehabilitation training to provide a new reference method of treatment for CAI patients.

## 2. Materials and Methods

### 2.1. Study Design

The study was a single-blind randomized controlled trial for the intervention method for subjects. The CONSORT 2010 statement [33] was followed in the design of this randomized controlled trial. The Ethics Committee of Chengdu Sport University approved this study (Ref. No: 2022-43), which complied with the Declaration of Helsinki.

### 2.2. Subjects

Subjects were recruited at Guangxi Normal University through questionnaires and posters. All the subjects signed a consent form before participating in the study.

Inclusion and exclusion criteria conformed to the 2019 consensus statement and recommendations made by the International Ankle Consortium [5].

Inclusion criteria:(1)Subjects had at least one history of ankle sprain in the past 12 months, causing pain and swelling, and the time to lose normal function within 1 day or more;(2)The affected ankle of subjects felt “soft leg”, and/or repeatedly sprains and/or “unstable”;(3)The Cumberland ankle instability tool (CAIT) scores for the subjects were less than or equal to 27.

Exclusion criteria:(1)Subjects had undergone surgery on any lower limb musculoskeletal structure in the past (i.e., bone, joint structure, nerve);(2)In the past three months, the subject suffered an acute injury (such as a sprain, or fracture) to the musculoskeletal structure of other joints of the lower limb, resulting in at least one day of required physical activity interruption;(3)Subjects had diseases such as the nervous system and vestibular system;(4)Subjects had other nerve injuries that could affect balance and muscle strength;(5)Subjects had accepted any other type of treatment during the trial.

Subjects who were incorrectly included because they did not fulfill the inclusion criteria, those who met the inclusion criterion but did not cooperate with treatment after inclusion, and those who could not participate on time were included in the rejection and abscission criteria.

### 2.3. Sample Size

The sample size was calculated by the mixed model analysis software (G*power 3.1, Düsseldorf, Germany) through repeated measurement ANOVA. The effect size was estimated to be 0.3 with 0.8 power and an α value of 0.05. According to the sample size calculation, at least 39 subjects had to be recruited. A total of 48 subjects (16 per group), taking subject loss into account, had to be recruited.

### 2.4. Measurements

#### 2.4.1. Star Excursion Balance Test (SEBT)

SEBT is a standard measurement method for dynamic balance, which also can be conducted to evaluate the dynamic postural control disorder induced by musculoskeletal injuries (such as chronic ankle instability) and assess the risk of injury of lower limbs [34]. The SEBT simple version was conducted in this study. There were three test directions consisting of anterior (A), posteromedial (PM), and posterolateral (PL). During the test, subjects stood on the star chart with the test foot, and the non-test foot, respectively, and sequentially stretched in three directions as far as possible to touch the maximum distance. If the test foot moved, rotated, or lost balance during the test, we would remeasure it. We calculated the average value based on the outcomes of three successful tests (M1, M2, M3). In addition, the results need to be standardized (SEBTC%) by dividing by the leg length of each test foot of the subjects because the subjects’ legs varied in length. The leg length (L) was measured by tape measures from the anterior superior iliac spine to the medial ankle. The calculation formula [31,35] is as follows
(ICC = 0.98–0.99):
SEBT (A, PM, PL)% = ((M1 + M2 + M3)/3)/L × 100%);
SEBTC% = (SEBT-A% + SEBT-PM% + SEBT-PL%)/3).

#### 2.4.2. Weight-Bearing Dorsiflexion Range of Motion (WB-DFROM)

WB-DFROM was conducted to evaluate the ankle dorsiflexion motion angle of subjects. WB-DFROM was used to simulate the function position of ankle dorsiflexion in daily physical activity [36]. Before the test, the researcher marked the wall and the perpendicular floor with colored tape. During the test, subjects faced the wall and stood with their feet forward and backward with the test foot stepped on the tape marked on the floor, pointed their toes vertically to the wall, and keep the longitudinal axis of the test foot in the center of the tape. Then the subjects bowed their knees toward the wall, slowly moved their leg back when the knee just came in contact with the wall and bowed their knees toward the wall again. Repeat this process until the heel was about to lift. Researchers marked the position where the big toe stepped on the tape with a marking pen, then measured the vertical distance between the marking point and the wall with a ruler, measuring three times, and obtain the best result [31,37] (ICC = 0.80–0.99) [38].

#### 2.4.3. Muscle Strength

The muscle around the ankle of CAI patients would be wasting and weak, increasing the risk of reinjury [39]. The assessment of muscle strength around the ankle of patients was conducive to the design and improve the rehabilitation plan [40,41]. The muscle strength was measured by the hand-held dynamometer (Hoggan Mi-croFet2 muscle strength tester). Subjects were required to lie in the supine position with the ankle of the test foot stretched out to the bedside, to test the muscle strength of the ankle dorsiflexion, plantar flexion, varus, and eversion. Each direction was measured five times and taken the maximum value (ICC = 0.83–0.94) [42]. The details of the measurement are presented in Table 1.

#### 2.4.4. Range of Movement (ROM)

A sufficient ankle range of movement is necessary for daily physical activity such as walking and running, but the ankle range of movement of patients with CAI was variably limited. Therefore, the primary physical treatment goal was to improve the ankle range of movement [43,44]. Ankle range of movement was measured for dorsal flexion, plantar flexion, varus, and eversion. It was measured with the joint range of movement measuring ruler. The maximum pain-free active range of movement was recorded for each movement. Three measurements were taken for each movement and the best result would be selected (ICC = 0.68–0.88) [45,46].

(1)Measurement of the ankle dorsiflexion and plantar flexion range of movement: The subject was in the prone position, and the ankle was in the neutral position. The axis was located about 2.5 cm below the midpoint of the ankle. The fixed leg was parallel to the long axis of the fibula and the moving leg was parallel to the fifth metatarsal bone.(2)Measurement of the ankle varus and eversion range of movement: The subject was in the prone position, and the ankle was in the neutral position. The axis was located near the outer side of the calcaneus. The fixed leg was parallel to the long axis of the tibia and the moving leg was parallel to the plantar surface of the heel.

Since the included angle between the fixed leg and the moving leg was 90° when measuring the starting position, it was necessary to deduct 90° after the measurement to obtain the right range of movement.

#### 2.4.5. Visual Analog Scale (VAS)

The self-comfort visual analog scale (SCS-VAS) and the self-induced stability scale (SISS-VAS) for the ankle of the subjects were measured. The VAS was conducted by a 10 cm long linear visual analog scale with numbers ranging from 0 to 10. 0 represented the highest degree of comfort or stability, while 10 represented the lowest degree of either. From 0 to 10, comfort or stability decreased in turn [47,48] (ICC = 0.71–0.99).

#### 2.4.6. Cumberland Ankle Instability Tool (CAIT)

CAIT, which consists of 9 questions and involves the self-perception of ankle stability in daily life [49], was created to evaluate the stability of the ankle. CAIT has shown a good intraclass correlation coefficient (ICC = 0.96) [50] with high diagnostic sensitivity and specificity [51].

### 2.5. Interventions

Subjects were randomly divided into the experimental group (EG), the control group (CG), and the sham group (SG) by using a simple random assignment sequence generated by Stata 12.0 software (www.stata.com (accessed on 15 March 2022), StataCorp, TX, USA). The proportion of subjects in each group was 1:1:1. Before the intervention, the paper written with the interventions was placed in sealed opaque envelopes in a 1:1:1 ratio by an independent researcher (not involved in the intervention), and subjects randomly drew envelopes to determine the received interventions. Subjects were required not to discuss the details of the intervention with the researchers. All groups received three treatment sessions per week for four weeks. For CG, routine rehabilitation training for the ankle was conducted. For EG, routine rehabilitation training combined with Maitland mobilization for the ankle was conducted. For SG, routine rehabilitation training combined with sham manual therapy for the ankle was conducted. The above treatment was carried out by a physiotherapist with more than three years of experience.

#### 2.5.1. Balance Training

Subjects were conducted stable plane and unstable plane balance training, respectively. The stable plane training was carried out on flat ground, and the unstable plane training was carried out on the balance pad. The training was divided into two types, one was single-leg standing with eyes open and the other one was single-leg standing with eyes closed [52].

(1)Training with eyes open: Subjects were required to keep their bodies upright, abduct their upper limbs 90 degrees, lift the healthy lower limbs to the knee of the affected side, keep the inner side of the healthy ankle at the same level as the healthy knee, and keep their bodies stable by looking straight ahead for one minute.(2)Training with eyes closed: Subjects were required to keep their body balance in advance, then close their eyes. The rest of the training was the same as training with eyes open.

Subjects were required to repeat each training 3 times with a 10-s break in between each repetition.

#### 2.5.2. Muscle Strength Rehabilitation Training

Use an elastic band to conduct plantar flexion, dorsiflexion, varus, and eversion resistance training in the neutral position of the ankle, varus and eversion resistance training in the plantar flexion position of the ankle, and varus and eversion resistance training in the dorsiflexion position of ankle. Subjects were required to reach the maximum painless joint range of movement while conducting the training. Repeat 8 times per group in each direction, rest for 3 min during the repetition and repeat 3 groups per training [14].

#### 2.5.3. Manual Therapy

The physiotherapist conducted Maitland mobilization for manual therapy [53,54].

(1)Talocrural joint longitudinal traction: Subjects lay in the supine position with the heel at the treatment bedside. The physiotherapist conducted the calcaneus’s level III traction action relative to the distal leg’s long axis.(2)Subtalar joint forward/backward sliding: Subjects lay in the supine position with the heel at the treatment bedside. The physiotherapist placed one hand on the instep and conducted the level I traction. Moreover, the other hand was placed on the posterior distal calcaneus. Then the physiotherapist conducted the level III forward/backward sliding motion of the calcaneus relative to the talus.(3)Subtalar joint Inside/outside sliding: Subjects lay in the prone position or the lateral decubitus position with the ankle propped up by towel rolls at the treatment bedside. The physiotherapist stabilized the talus of the subject with one hand, then placed the other hand’s palm on the medial calcaneus then conducted level III outside sliding. Or the physiotherapist placed the other hand’s palm on the lateral calcaneus and then conducted level III inside sliding.

Each group received 30 s of manual therapy, with 1 min rest between groups. There were 3 groups in total.

#### 2.5.4. Sham Manual Therapy

The sham manual therapy was conducted in the same position as the manual therapy. The only difference between them was that sham manual therapy did not involve any type of exercise, but the physiotherapist maintained hand contact with the skin for a period of ten minutes.

### 2.6. Statistical Analysis

The Shapiro–Wilk test was used to evaluate the normality of variables. The mean and standard deviation (SD) were reported for the descriptive analysis of the quantitative variables. For the categorical variable, chi-square tests were reported. The paired sample T-test was used for intragroup statistical analysis. After the intervention, one-way repeated measures ANOVA was applied to evaluate whether the muscle strength around the ankle, ankle range of movement, and lower limb balance ability were statistically significant. Moreover, the post hoc multiple comparisons were conducted through Bonferroni. The effect size was calculated through the partial Eta square (η^2^), 0.01 ≤ η^2^ < 0.06 represented the small effect size, 0.06 ≤ η^2^ < 0.14 represented the medium effect size and η^2^ ≥ 0.14 represented the large effect size.

All the analyses were conducted with SPSS v.22 software (SPSS Inc., Chicago, IL, USA), *p*-values < 0.05 were considered statistically significant and confidence intervals (CIs) were 95% [30,55].

## 3. Results

A total of 52 subjects were recruited. A total of 4 subjects were excluded, 2 of whom had bilateral ankle instability and the other 2 had a history of ankle surgery. Finally, 48 subjects (35 males and 13 females) were selected. During the research, 2 subjects dropped out at the follow-up and 1 subject dropped out due to an ankle sprain. 45 subjects completed the study, of which 16 were allocated to the EG group, 15 to the CG group, and 14 to the SG group. A flow diagram of subjects is presented in Figure 1.

### 3.1. General Information of Subjects

The general information of subjects: age = 20.33 ± 1.08 (y), height = 1.73 ± 0.08 (m), weight = 67.05 ± 10.75 (kg), BMI = 22.23 ± 2.77 (kg/m^2^), CAIT = 16.13 ± 2.86 (score). There was no statistically significant difference in the general information of the three group subjects (*p* > 0.05). The details are presented in Table 2.

### 3.2. CAIT, SIS-VAS, SCS-VAS, and WB-DFROM (cm)

The study showed that there was no statistically significant difference among the indexes of each group before intervention (*p* > 0.05). After 4 weeks of intervention, the scores of CAIT, SCS-VAS, and SISS-VAS in each group increased, but there was no statistically significant difference between EG and CG, SG (*p* > 0.05). The details are presented in Table 3.

### 3.3. SEBT

In terms of SEBT, after 4 weeks of intervention, there was a statistically significant difference among the three groups: SEBT-A (*p* < 0.001, F = 17.453, η^2^ = 0.573), SEBT-PM (*p* = 0.001, F = 8.584, η^2^ = 0.398), SEBT-PL (*p* = 0.006, F = 6.324, η^2^ = 0.327), SEBT-C (*p* = 0.002, F = 7.803, η^2^ = 0.375). The post hoc multiple comparisons showed that: there was a statistically significant difference between EG and CG in terms of SEBT-A (*p* = 0.006, 95%CI = 2.388–14.265) and SEBT-PM (*p* = 0.042, 95%CI = 0.288–16.712), and there was a statistically significant difference between EG and SG in terms of SEBT-A (*p* = 0.004, 95%CI = 2.838–14.348), SEBT-PM (*p* = 0.004, 95%CI = 2.838–14.348), SEBT-PL (*p* = 0.002, 95%CI = 3.236–14.053), SEBT-C (*p* = 0.003, 95%CI = 3.229–15.354) (Figure 2A–D).

### 3.4. Range of Movement

After 4 weeks of intervention, the ankle range of movement in the three groups was improved compared with that before (*p* < 0.05) (Figure 3). There was a statistically significant difference in the dorsiflexion ankle range of movement among the three groups (*p* = 0.015, F = 5.063, η^2^ = 0.297). The post hoc multiple comparisons showed that there was a statistically significant difference between EG and CG (*p* = 0.003, 95%CI = 1.492–6.662) (Figure 3A). There was no statistically significant difference in the plantar flexion ankle range of movement among the three groups. The post hoc multiple comparisons showed that there was a statistically significant difference between EG and SG (*p* = 0.038, 95%CI = 0.416–16.507) (Figure 3B). Moreover, there was a statistically significant difference among the three groups in the varus range of movement (*p* = 0.044, F = 3.568, η^2^ = 0.229) (Figure 3B). Moreover, there was also a statistically significant difference in the eversion range of movement among the three groups (*p* = 0.029, F = 4.110, η^2^ = 0.255). Moreover, the post hoc multiple comparisons presented that there was a statistically significant difference between EG and CG in the eversion range of movement (*p* = 0.031, 95%CI = 0.634–14.289) (Figure 3D).

### 3.5. Muscle Strength

The results indicated that the muscle strength of all groups had an enhancement compared with that before the four weeks intervention (*p* < 0.05) (Figure 4). There was a statistically significant difference in the strength of dorsal flexor muscles among the three groups (*p* = 0.046, F = 3.473, η^2^ = 0.211). The post hoc multiple comparisons showed that there was a statistically significant difference between EG and CG in dorsal flexor force (*p* = 0.048, 95%CI = 0.026–8.860) (Figure 4A). There was no statistically significant difference in plantar flexor strength among the three groups (Figure 4B). There was a statistically significant difference in varus muscle strength among the three groups (*p* = 0.002, F = 7.890, η^2^ = 0.378). There was a statistically significant difference between EG and CG (*p* = 0.023, 95%CI = 0.350–5.021). Moreover, there was a statistically significant difference between EG and SG (*p* = 0.031, 95%CI = 0.213–4.872) (Figure 4C). There was a statistically significant difference in eversion muscle strength among the three groups (*p* = 0.046, F = 3.465, η^2^ = 0.210) (Figure 4D).

## 4. Discussion

According to statistics, about 71,200 people have ankle sprains every day [56]. The majority of people ignore it. About 33% of affected people did not receive systematic rehabilitation treatment after the first ankle sprain. In the following two years, they would continue to suffer from symptoms such as decreased ankle range of movement, decreased muscle strength, proprioception abnormality, and decreased excitability of the nervous system, which weakened lower limb function and raised the risk of re-injury [57,58]. After the first injury, the recurrence rate is as high as 80% [54]. That is why the main purpose of the intervention is to increase the ankle range of movement, improve the nerve conduction capacity, enhance muscle strength and improve dynamic balance control.

After 4 weeks of intervention, SEBT-A, SEBT-PM, SEBT-PL, and SEBT-C of EG, CG, and SG were significantly improved, and there was a statistical difference among the three groups. In terms of the improvement of SEBT-A, SEBT-PM, SEBT-PL, and SEBT-C, there was a statistically significant difference between EG and SG. This may be due to the Maitland mobilization improving the ankle range of movement. After routine rehabilitation, the balance of the ankle was also improved. In the comparison of EG and CG, there was a statistically significant difference in terms of SEBT-A and SEBT-PM. Muscle strength training can improve the proprioceptive acuity and balance of the ankle [59]. Moreover, the improvement of EG was most obvious, which could be seen that adding Maitland mobilization to routine rehabilitation training for CAI patients may have a more positive effect on balance. There are free nerve endings and mechanical receptors around the ligaments near the ankle which can perceive joint speed in movement. Due to the restricted flexibility of the ankle, abnormal signals from mechanical receptors will be introduced, affecting the proprioception of the ankle and, to varying degrees, a decline in balance [60,61]. The static position, speed, and direction of the ankle may change during Maitland mobilization at various frequencies. It functions as a mechanical receptor stimulator around the ankle, altering the potential of the muscles around the ankle and raising the excitability of the spinal dorsal horn of the muscle group motor units [62]. It may enhance the sensory information input from the talocrural joint and surrounding tissues. Moreover, the change of joint position can amplify the transmission of proprioception signal to the center, improve the ability of nerve transmission, and promote the central perception of ankle position and movement [63,64], all of which have the potential to improve the dynamic stability of the ankle and lower limbs.

The restricted ankle dorsiflexion range of movement is a common dysfunction of patients with CAI. The symptom is caused by forwarding displacement of the talus or forward displacement of the distal fibula relative to the tibia [20]. As a result, the talus is unable to slide backward effectively during movement, and the ankle joint is in an unstable closed state when landing [65]. In addition, the subtalar joint as a “mitered hinge” is also one of the crucial joints of the lower limbs, and the limitation of the subtalar joint is also a potential factor affecting ankle kinematics [66,67]. The movement of the subtalar joint usually occurs in three planes of the foot at the same time, which is called three-plane joint movement. During the movement, the ankle dorsiflexion and abduction were caused by the inter-connection of subtalar joint pronation and supination with calf rotation [68,69]. The ankle will be sprained if the interconnection is broken. Maximum dorsiflexion of the ankle joint causes the talus to rotate outward, which limits the talocrural joint’s ability to slide medially. Maximum dorsiflexion will reduce bone flexibility, which means a higher risk of ankle varus sprain [70]. In terms of the range of movement, the subjects’ ankle dorsiflexion, plantar flexion, varus, and eversion range of movement were significantly improved after 4 weeks of intervention, but there was no statistically significant difference in varus range of movement after increased manual therapy. However, the improvement of EG in its four directions range of movement was significantly better than that of the other two groups. It may be due to the Maitland mobilization for the talocrural joint and subtalar joint to make the talus slide backward relative to the fibula, correct the position of the talus in the articular pan, and improve the ankle joint range of movement [71]. Moreover, then the lower limb force line will be adjusted to restore the stable mechanical structure [20]. Sliding the joint in a specific direction can reduce compression of structural and synovial, to improve the movement of the ankle.

For the patient with CAI, both the reaction and activation times of the long and short fibular muscles in the affected lower limb are significantly prolonged [72]. Due to this, the patient repeatedly sprains their ankle while engaging in dynamic activities. The main muscle injuries caused by CAI are the weakness of ankle plantar flexor muscles (gastrocnemius, soleus), the weakness of eversion muscles (long and short fibular muscle), and the slowing of eversion response [14]. In this study, EG, CG, and SG were significantly improved after 4 weeks of intervention. Moreover, only the muscle strength for dorsiflexion and varus were statistically different between EG and the other two groups. The situation indicated that the muscle strength in all directions had been greatly improved after adding manual therapy to the routine rehabilitation training. It is suggested that manual therapy and routine rehabilitation training have a synergistic effect, which can more effectively improve muscle strength around the ankle and help to improve stability. Fisher B E et al. [73] found that joint mobilization had a different influence on corticospinal excitability. After the talocrural joint mobilization, the corticospinal excitability of the tibialis anterior muscle was increased. However, in a previous study [53], researchers conducted joint mobilization combined neuromuscular training 8 times over 4 weeks to 15 CAI patients. The joint mobilization included traction in the longitudinal direction to the talocrural joint, grade III and grade IV anterior to posterior mobilization to the talus, and grade III and grade IV mobilization to the distal tibiofibular joint. They resulted that the intervention had no effect on the activation of muscles around the ankle, and had no obvious help in the improvement of muscle strength. It is inconsistent with our research results. In our study, 12 times over 4 weeks of Maitland mobilization, including talocrural joint longitudinal traction, grade III subtalar joint forward/backward sliding, and grade III subtalar joint inside/outside sliding, combined with routine rehabilitation was performed. Therefore, the difference may be caused by the species and grade of the mobilization, and the frequency of the intervention. Our study did not measure some valuable physiological indicators such as EMG because of the limited experimental condition. So, further investigation and verification are necessary.

Regular self-report of ankle severity for CAI patients is conducive to the implementation and improvement of rehabilitation strategies [74]. After 4 weeks of intervention, the scores of CAIT, SCS-VAS, and SISS-VAS of EG, CG, and SG were improved, but there was no statistical difference among the three groups. Surprisingly, when only conducting routine rehabilitation training for subjects, a positive impact on the self-report of the severity of ankle instability would occur. However, when sham manual therapy was added, the impact was weakened. In this regard, perhaps we should increase the sample size to solve this problem in the future.

## 5. Limitations

Three of the subjects withdrew from the study, and the number of men and women was imbalanced, which may affect the statistical efficacy. Moreover, the study only used a single-blind design for the participants rather than conducting a double-blind study. Moreover, better comparative analysis cannot be achieved due to the lack of a control group with simple manual therapy. Additionally, the hand-held dynamometer used for the muscle strength tests may have introduced human error into the process, skewing the results. Therefore, in future research, objective instruments such as electromyograms and isokinetic dynamometers are supposed to be used in the muscle strength test to improve experimental rigor. Moreover, it is suggested that force plates and stereophotogrammetry are supposed to be conducted in the tests for balance and joint range of movement.

## 6. Conclusions

In summary, the results suggest that the balance, ankle range of movement, and muscle strength of patients with CAI were improved after three types of intervention methods. Compared with routine rehabilitation training alone, routine rehabilitation training combined with Maitland mobilization seems to improve the balance ability and ankle range of movement of patients with CAI, but it is not obvious enough to promote the improvement of muscle strength. It is necessary to increase the mid-term and long-term follow-up survey to clarify the effectiveness of Maitland mobilization in CAI in future research.

## Figures and Tables

**Figure 1 ijerph-19-15328-f001:**
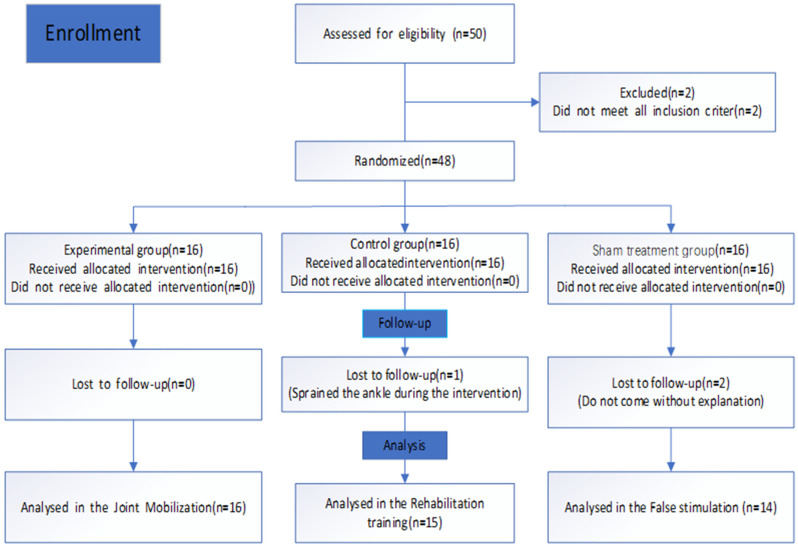
Flow diagram.

**Figure 2 ijerph-19-15328-f002:**
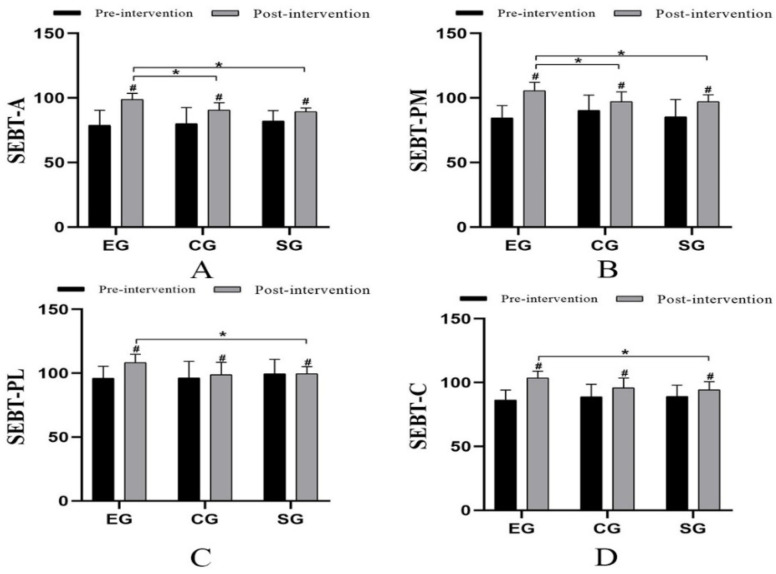
Comparison of SEBT (cm) values among the three groups. # Comparison between groups; * Significant when compared with the EG group (*p* < 0.05). (**A**): SEBT-A; (**B**): SEBT-PM; (**C**): SEBT-PL; (**D**): SEBT-C.

**Figure 3 ijerph-19-15328-f003:**
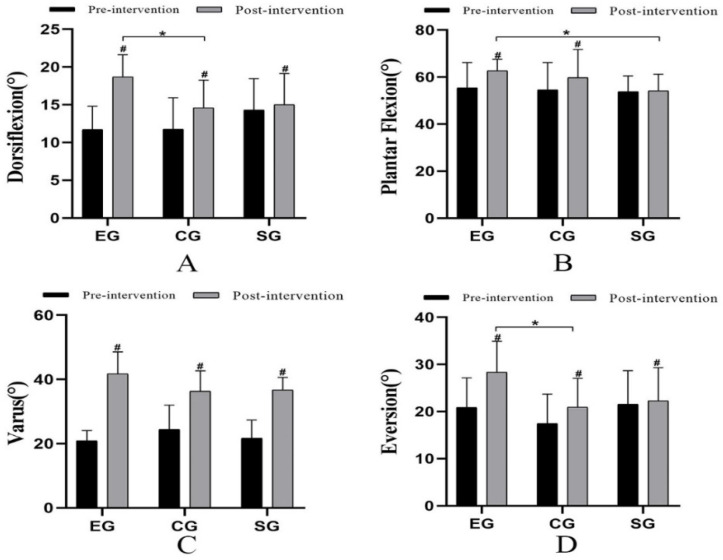
Comparison of range of movement (°) among the three groups. # Comparison between groups; * Significant when compared with the EG group (*p* < 0.05). ((**A**): Dorsiflexion ankle range of movement; (**B**): Plantar flexion ankle range of movement; (**C**): Varus ankle range of movement; (**D**): Eversion ankle range of movement).

**Figure 4 ijerph-19-15328-f004:**
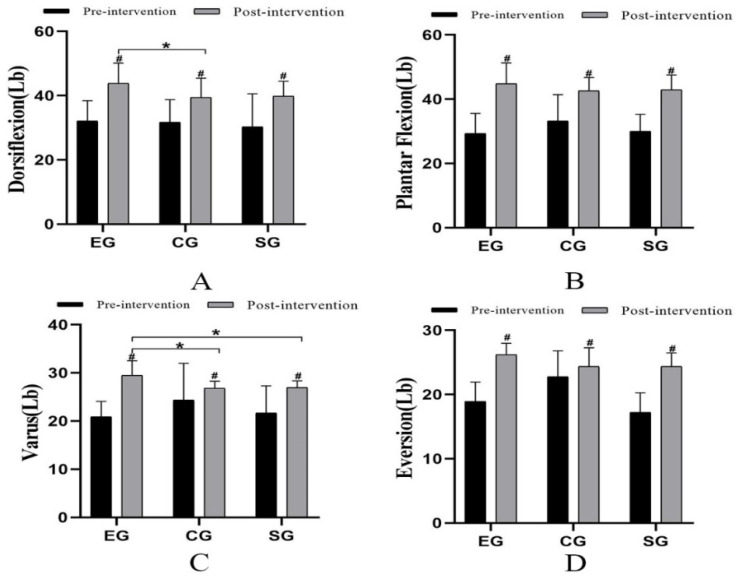
Comparison of muscle strength (Lb) among the three groups. # Comparison between groups; * Significant when compared with the EG group (*p* < 0.05). ((**A**): Ankle dorsiflexor strength; (**B**): Ankle plantar flexion strength; (**C**): Ankle varus strength; (**D**): Ankle eversion strength).

**Table 1 ijerph-19-15328-t001:** The details of the measurement.

Direction of Movement	Position of Subject	Position of Joint	Position of HHD	Position of Conductor
Dorsiflexion	Position of supine	Neutral position	1st, 2nd, and 3rd metatarsal bones of the forefoot	The opposite side of the subject, leaning backward
Plantar flexion			1st, 2nd, and 3rd metatarsal bones of the front instep	
Varus			Outside of the 1st metatarsal bone	On the side of the subject, push backward
eversion			Outside of the 5th metatarsal bone	

Note: Hand-held dynamometer, HHD.

**Table 2 ijerph-19-15328-t002:** Descriptive data of subjects in each group. Data are reported as mean ± SD.

Variable	Group			*p* Value
	EG (n = 16)	CG (n = 15)	SG (n = 14)	
Characteristic	No. (%)			
Sex	0.961			
Male	11 (69)	11 (73)	10 (71)	
Female	5 (31)	4 (27)	4 (23)	
Ankle	0.716			
Left	4 (25)	5 (33)	8 (57)	
Right	12 (75)	10 (67)	6 (43)	
	Mean ± SD			
Age, y	20.00 ± 1.58	20.31 ± 0.75	20.69 ± 0.63	0.271
Height, m	1.72 ± 0.08	1.74 ± 0.08	1.74 ± 0.08	0.713
Weight, kg	64.65 ± 7.66	65.12 ± 7.90	71.38 ± 14.63	0.207
BMI, kg/m^2^	21.80 ± 1.72	21.40 ± 2.44	23.48 ± 3.57	0.126
CAIT score	15.62 ± 3.28	17.31 ± 2.53	15.46 ± 2.54	0.191

Note: CAIT, Cumberland Ankle Instability Tool. BMI: body mass index. SD: standard deviation.

**Table 3 ijerph-19-15328-t003:** CAIT, SIS-VAS, SCS-VAS, and WB-DFROM before and after the intervention. Data are reported as mean ± SD.

Variable	Group								
	EG (n = 16)		CG (n = 15)		SG (n = 14)		F Values	*p* Values	η^2^
	Pre-Intervention	Post-Intervention	Pre-Intervention	Post-Intervention	Pre-Intervention	Post-Intervention			
CAIT	15.62 ± 3.28	20.14 ± 2.76 ^#^	17.31 ± 2.53	19.85 ± 2.34 ^#^	15.46 ± 2.54	18.57 ± 2.66 ^#^	1.474	0.248	0.102
SISS-VAS	2.15 ± 0.80	4.30 ± 0.94 ^#^	1.62 ± 0.65	4.23 ± 0.59 ^#^	1.38 ± 0.51	3.92 ± 1.03 ^#^	0.708	0.503	0.056
SCS-VAS	1.85 ± 0.90	4.61 ± 0.87 ^#^	1.92 ± 0.76	4.61 ± 0.65 ^#^	1.54 ± 0.52	4.15 ± 0.80 ^#^	1.220	0.313	0.092
WB-DFROM (cm)	9.58 ± 2.60	15.15 ± 1.57 ^#^	11.12 ± 2.94	12.35 ± 2.58 ^#^*	11.65 ± 3.40	12.12 ± 3.00 *^#^	6.90	<0.01	0.36

Note: SISS-VAS, self-induced stability scale; SCS-VAS, self-comfort scale; WB-DFROM, weight-bearing dorsiflexion range of motion; ^#^ comparison between groups; * significant when compared with the EG group; (*p* < 0.05).

## Data Availability

The data are not publicly available for privacy or ethical reasons.

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
