# Peer review of "Effectiveness of the Rehabilitation Training Combined with Maitland Mobilization for the Treatment of Chronic Ankle Instability: A Randomized Controlled Trial"

_ijerph, 2022, doi:10.3390/ijerph192215328_

Round 1

Reviewer 1 Report

Dear authors

The abstract adequately contains the basic elements, of the IMRaD format

Concise and clear presentation of the problem under study.

For the reader, a more explicit and articulated expression of the research objectives is recommended.

 An explicit structured writing of the conclusions is recommended as a synoptic response to the objectives.

Author Response

Dear editor,

We deeply appreciate your valuable comments concerning our manuscript. The suggestions are all very helpful for the revision and improvement. We have carefully studied these comments and made related modifications, which we wish could meet with your approval.

1.The abstract adequately contains the basic elements, of the IMRaD format, Concise and clear presentation of the problem under study. For the reader, a more explicit and articulated expression of the research objectives is recommended.

Response:

Thank you for this suggestion.

Abstract

The study aims to determine whether if routine rehabilitation training combined with Maitland mobilization is more effective than routine rehabilitation training alone in patients with chronic ankle instability, intending to provide a novel rehabilitation strategy for chronic ankle instability.48 subjects were divided into three groups: EG(Maitland mobilization and routine rehabilitation), CG(routine rehabilitation) and SG(sham mobilization and routine rehabilitation). The intervention was performed three times each week for 4 weeks, for a total of 12 sessions. Before and after the intervention, the muscle strength, star excursion balance test (SEBT), weight-bearing dorsiflexion range of motion (WB-DFROM), ankle range of movement, Cumberland ankle instability tool (CAIT), self-comfort visual analog scale (SCS-VAS), and self-induced stability scale (SISS-VAS) were assessed. The results showed that the improvement of SEBT, WB-DFROM and active ankle range of movement without the pain in EG was more obvious than CG and SG, but the improvement of the self-report of ankle severity and muscle strength was not. Compared with routine rehabilitation training alone, routine rehabilitation training combined with Maitland mobilization for patients with chronic ankle instability may more benefit in terms of balance, ankle range of movement than routine rehabilitation alone, but the improvement in muscle strength was not evident enough.

  1. An explicit structured writing of the conclusions is recommended as a synoptic response to the objectives.

Response:

Thank you for this suggestion.

In summary, the results showed that the balance, ankle range of movement and muscle strength of patients with CAI were improved after three types of intervention methods. Compared with routine rehabilitation training alone, routine rehabilitation training combined with Maitland mobilization can effectively improve the balance ability and ankle range of movement of patients with CAI, but it is not obvious enough to promote the improvement of muscle strength. It is necessary to increase the mid-term and long-term follow-up survey to clarify the effectiveness of Maitland mobilization in CAI in the future research.

Reviewer 2 Report

In this interesting work, Yin and colleagues investigated the effects of routine rehabilitation combined with Maitland mobilization in patients suffering from chronic ankle instability.

My main doubt about this article concerns evaluation methods. For instance, an instrumental assessment of balance (using a force plate) would have been crucial in such a work. The limitations section speaks of the possible human error introduced by the use of the dynamometer but I think that this risk is substantially present in all the evaluation methods proposed, which are mainly based on tests and questionnaires. This risks devaluing the results of the study which, in itself, is well structured. I have the below additional minor points:

- line 330-333:  “In the following two years, they would 330 continue to suffer from symptoms such as decreased ankle range of movement, decreased 331 muscle strength, proprioception abnormality and decreased excitability of the nervous 332 system, which weakened lower limb function and raised the risk of re-injury.”

Add at least 2 references

- line 340-342: “As a result, it is clear that adding Maitland mobilization to the routine rehabilitation training of CAI has a more beneficial impact on the improvement of balance.”

For the reasons mentioned above, although the results reported seem to confirm the proposed thesis, I would recommend a less absolutist statement.

- line 359-361: “In addition, the subtalar joint as a “mitered hinge” is 359 also one of the crucial joints of the lower limbs, and the limitation of the subtalar joint is 360 also a potential factor affecting ankle kinematics.”

 Please add some references here

- line 390-394: ”However, previous studies have found that manual therapy combined with neuromuscular training has no effect on the activation of muscles around the ankle, and has no obvious help in the improvement of muscle strength [48]. It is inconsistent with our research results, which still need further investigation and verification.”

This deserves to be discussed better

Author Response

Dear editor,

We deeply appreciate your valuable comments concerning our manuscript. The suggestions are all very helpful for the revision and improvement. We have carefully studied these comments and made related modifications, which we wish could meet with your approval.

1.- line 330-333: “In the following two years, they would 330 continue to suffer from symptoms such as decreased ankle range of movement, decreased 331 muscle strength, proprioception abnormality and decreased excitability of the nervous 332 system, which weakened lower limb function and raised the risk of re-injury.”——Add at least 2 references.

Response:

Thank you for this suggestion.

In the following two years, they would continue to suffer from symptoms such as de-creased ankle range of movement, decreased muscle strength, proprioception abnormality and decreased excitability of the nervous system, which weakened lower limb function and raised the risk of re-injury[57,58]. After the first injury, the recurrence rate is as high as 80%[54].

Add 2 references:

57.Al Adal, S.; Pourkazemi, F.; Mackey, M.; Hiller, C.E. The Prevalence of Pain in People With Chronic Ankle Instability: A Systematic Review. Journal of athletic training 2019, 54, 662-670, doi:10.4085/1062-6050-531-17.

58.Bestwick-Stevenson, T.; Wyatt, L.A.; Palmer, D.; Ching, A.; Kerslake, R.; Coffey, F.; Batt, M.E.; Scammell, B.E. Incidence and risk factors for poor ankle functional recovery, and the development and progression of posttraumatic ankle osteoarthritis after significant ankle ligament injury (SALI): the SALI cohort study protocol. BMC musculoskeletal disorders 2021, 22, 362, doi:10.1186/s12891-021-04230-8.

2.- line 340-342: “As a result, it is clear that adding Maitland mobilization to the routine rehabilitation training of CAI has a more beneficial impact on the improvement of balance.”

For the reasons mentioned above, although the results reported seem to confirm the proposed thesis, I would recommend a less absolutist statement.

Response:

Thank you for this suggestion.

And, the improvement of EG was most obvious, which could be seen that adding Mait-land mobilization to routine rehabilitation training for CAI patients may have a more pos-itive effect on balance.

3.- - line 359-361: “In addition, the subtalar joint as a “mitered hinge” is 359 also one of the crucial joints of the lower limbs, and the limitation of the subtalar joint is 360 also a potential factor affecting ankle kinematics.”——Please add some references here.

Response:

Thank you for this suggestion.

In addition, the subtalar joint as a “mitered hinge” is also one of the crucial joints of the lower limbs, and the limitation of the subtalar joint is also a potential factor affecting ankle kinematics[66,67]. The movement of the subtalar joint usually occurs in three planes of the foot at the same time, which is called three-plane joint movement. During the movement, the ankle dorsiflexion and abduction were caused by the inter-connection of subtalar joint pronation and supination with calf rotation [68,69].

Add 2 references:

66.Hertel, J.; Corbett, R.O. An Updated Model of Chronic Ankle Instability. Journal of athletic training 2019, 54, 572-588, doi:10.4085/1062-6050-344-18.

67.Almeida, M.O.; Davis, I.S.; Lopes, A.D. Biomechanical Differences of Foot-Strike Patterns During Running: A Systematic Review With Meta-analysis. The Journal of orthopaedic and sports physical therapy 2015, 45, 738-755, doi:10.2519/jospt.2015.6019.

4.- line 390-394: ”However, previous studies have found that manual therapy combined with neuromuscular training has no effect on the activation of muscles around the ankle, and has no obvious help in the improvement of muscle strength [48]. It is inconsistent with our research results, which still need further investigation and verification.”——This deserves to be discussed better

Fisher B E etc.[73] found that joint mobilization had a different influence on corticospinal excitability. After the talocrural joint mobilization, the corticospinal excitability of the tibialis anterior muscle was increased. However, in a previous study[53], researchers con-ducted joint mobilization combined neuromuscular training for 8 times over 4 weeks to 15 CAI patients. The joint mobilization included traction in the longitudinal direction to the talocrural joint, grade III and grade IV anterior to posterior mobilization to the talus and grade III and grade IV mobilization to the distal tibiofibular joint. They resulted that the intervention had no effect on the activation of muscles around the ankle, and had no obvious help in the improvement of muscle strength. It is inconsistent with our research results. In our study, 12 times over 4 weeks of Maitland mobilization, including talocrural joint longitudinal traction, grade III subtalar joint forward/backward sliding and grade III subtalar joint inside/outside sliding, combined with routine rehabilitation was performed. Therefore, the difference may be caused by the species and grade of the mobilization, and the frequency of the intervention. Our study did not measure some valuable physiological indicators such as EMG because of the limited experimental condition. So, further investigation and verification are necessary.

Add 1 references:

73.Fisher, B.E.; Piraino, A.; Lee, Y.Y.; Smith, J.A.; Johnson, S.; Davenport, T.E.; Kulig, K. The Effect of Velocity of Joint Mobilization on Corticospinal Excitability in Individuals With a History of Ankle Sprain. The Journal of orthopaedic and sports physical therapy 2016, 46, 562-570, doi:10.2519/jospt.2016.6602.

Reviewer 3 Report

Title: Effectiveness of the Rehabilitation Training Combined with Maitland Mobilization for the Treatment of Chronic Ankle Instability: a randomized controlled trial

This is a rigorously conducted clinical trial, in which they try to observe whether manual therapy together with rehabilitation can be more effective than other types of therapies. The following are the modifications that should be made.

Comments

Abstract: Please, reduce the length of the abstract and keep in mind that it should be in one paragraph. Make it clear what the main objective was.

Introduction:

An extensive introduction has been made in which the problems related to ankle sprain, its symptoms and the main limitations of the current treatments performed to reduce pain and increase proprioception are clearly explained.

Please, include:

1)    Line 65-73: Only one of the citations used (Holland et al.) is a clinical trial related to Maitland mobilization. Include more sources to support your work.

2)     Line 76: It is not clear what the primary objective of the clinical trial is. Please state a specific objective or hypothesis and, if necessary, secondary objectives.

Method:

1)    Line 82-84: The authors have not included the type of randomized clinical trial they have performed. Please describe it. Also indicate here (in addition to the limitations section) the type of blind used.

2)   The registration number of the clinical trial and the name of the site where it was registered have not been included.

3)  Line 97: Was it considered whether the subjects were taking any type of pain treatment? If it was not considered, this should be mentioned as a limitation.

4)     Line 180: What type of program was used to perform randomization and allocation?

5)     Please substantiate each intervention carried out in the study using the appropriate literature.

Results

1)    The groups have a serious imbalance between the number of men and women, why is this? If the study is based on the general population, the number should be equal. This should be named in the "limitations" section.

2)   Line 277-289: The results are continuously repeated, changing only the numbers. Please describe them consistently.

Discussion

1)   Line 318. The article does not match the statement. Please add an epidemiology article for this data.

2)    Line 340. Even if the results are favorable, one cannot make such an extreme claim that the intervention has a "more beneficial" impact.

Conclusion

Please include in your conclusions the relationship with the Maitland mobilization.

Author Response

Dear editor,

We deeply appreciate your valuable comments concerning our manuscript. The suggestions are all very helpful for the revision and improvement. We have carefully studied these comments and made related modifications, which we wish could meet with your approval.

  1. Line 65-73: Only one of the citations used (Holland et al.) is a clinical trial related to Maitland mobilization. Include more sources to support your work.

Thank you for this suggestion.

In previous studies on musculoskeletal diseases[25-27], manual therapy has been shown to have a positive recovery effect and is regarded as a good way to make the patients com-fortable by increasing the joint range of movement, stimulating nerve conduction path-ways, and improving proprioception [28-31].

Add 3 references:

  1. Hoch, M.C.; McKeon, P.O. Joint mobilization improves spatiotemporal postural control and range of motion in those with chronic ankle instability. Journal of orthopaedic research : official publication of the Orthopaedic Research Society 2011, 29, 326-332, doi:10.1002/jor.21256.
  2. Dogan, N.; Sahbaz, T.; Diracoglu, D. Effects of mobilization treatment on sacroiliac joint dysfunction syndrome. Revista da Associacao Medica Brasileira (1992) 2021, 67, 1003-1009, doi:10.1590/1806-9282.20210436.
  3. Rahbar, M.; Ranjbar Kiyakalayeh, S.; Mirzajani, R.; Eftekharsadat, B.; Dolatkhah, N. Effectiveness of acromioclavicular joint mobilization and physical therapy vs physical therapy alone in patients with frozen shoulder: A randomized clinical trial. Clinical rehabilitation 2022, 36, 669-682, doi:10.1177/02692155211070451.
  4. Line 76: It is not clear what the primary objective of the clinical trial is. Please state a specific objective or hypothesis and, if necessary, secondary objectives.

Thank you for this suggestion.

The objective of the study was as follows: aims to explore whether routine rehabilitation training combined Maitland mobilization for CAI patients has more positive improvement than routine rehabilitation alone, and to determine the effectiveness of Maitland mobilization for CAI patients. We hypothesized that the combination of routine rehabilitation training and Maitland mobilization would have a positive effect on balance, ankle range of movement, and muscle strength in patients with CAI, to explore whether Maitland mobilization is synergistic with routine rehabilitation training to provide a new reference method of treatment for CAI patients.

  1. Line 82-84: The authors have not included the type of randomized clinical trial they have performed. Please describe it. Also indicate here (in addition to the limitations section) the type of blind used.

Thank you for this suggestion.

The study was a single-blind randomized controlled trial for the intervention method for subjects. The CONSORT 2010 statement [33] was followed in the design of this ran-domized controlled trial. The Ethics Committee of Chengdu Sport University approved this study (Ref.No:2022-43), which complied with the Declaration of Helsinki.

  1. The registration number of the clinical trial and the name of the site where it was registered have not been included.

Thank you for this suggestion. We did not sample any tissues, organs or blood in the human body during the trial, so the register of the clinical trial was not conducted.

  1. Line 97: Was it considered whether the subjects were taking any type of pain treatment? If it was not considered, this should be mentioned as a limitation.

Thank you for this suggestion.

Exclusion criteria:

1) Subjects had undergone surgery on any lower limb musculoskeletal structure in the past (i.e., bone, joint structure, nerve);

2) In the past three months, the subject suffered an acute injury (such as a sprain, or fracture) to the musculoskeletal structure of other joints of the lower limb, resulting in at least one day of required physical activity interruption;

3) Subjects had diseases such as the nervous system and vestibular system;

4) Subjects had other nerve injuries that could affect balance and muscle strength.

5) Subjects had accepted any other type of treatment during the trial.

  1. Line 180: What type of program was used to perform randomization and allocation

Thank you for this suggestion.

Subjects were randomly divided into the experimental group (EG), the control group (CG), and the sham group (SG) by using a simple random assignment sequence generated by Stata12.0 software (www.stata.com, USA). The proportion of subjects in each group was 1:1:1. Before the intervention, the paper written with the interventions was placed in sealed opaque envelopes in a 1:1:1 ratio by an independent researcher (not involved in the intervention), and subjects randomly drew envelopes to determine the received interventions. Subjects were required not to discuss the details of the intervention with the researchers. All groups received three treatment sessions per week for four weeks. For CG, routine rehabilitation training for the ankle was conducted. For EG, routine rehabilitation training combined with Maitland mobilization for the ankle was conducted. For SG, routine rehabilitation training combined with sham manual therapy for the ankle was conducted. The above treatment was carried out by a physiotherapist with more than three years of experience.

  1. Please substantiate each intervention carried out in the study using the appropriate literature.

Thank you for this suggestion.

2.5.1.Balance Training[52]

Subjects were conducted stable plane and unstable plane balance training respec-tively. The stable plane training was carried out on the flat ground, and the unstable plane training was carried out on the balance pad. The training was divided into two types, one was single-leg standing with eyes open and the other one was single-leg standing with eyes closed.

1) Training with eyes open: Subjects were required to keep their bodies upright, ab-duct their upper limbs 90 degrees, lift the healthy lower limbs to the knee of the affected side, keep the inner side of the healthy ankle at the same level as the healthy knee, and keep their bodies stable by looking straight ahead for one minute.

2) Training with eyes closed: Subjects were required to keep their body balance in advance, then close their eyes. The rest of the training was the same as training with eyes open.

Subjects were required to repeat each training 3 times with a 10-second break in be-tween each repetition.

2.5.2. Muscle Strength Rehabilitation Training[14]

Use an elastic band to conduct plantar flexion, dorsiflexion, varus and eversion re-sistance training in the neutral position of the ankle, varus and eversion resistance train-ing in the plantar flexion position of the ankle, and varus and eversion resistance training in the dorsiflexion position of ankle. Subjects were required to reach the maximum pain-less joint range of movement while conducting the training. Repeat 8 times per group in each direction, rest for 3 minutes during the repetition and repeat 3 groups per training.

2.5.3. Manual Therapy[53,54]

The physiotherapist conducted Maitland mobilization for manual therapy.

1) Talocrural joint longitudinal traction: Subjects lay in the supine position with the heel at the treatment bedside. The physiotherapist conducted the calcaneus's level III trac-tion action relative to the distal leg's long axis.

2) Subtalar joint forward/backward sliding: Subjects lay in the supine position with the heel at the treatment bedside. The physiotherapist placed one hand on the instep and conducted the levelâ… traction. And the other hand was placed on the posterior distal cal-caneus. Then the physiotherapist conducted the level III forward/backward sliding motion of the calcaneus relative to the talus.

3) Subtalar joint Inside/outside sliding: Subjects lay in the prone position or the lateral decubitus position with the ankle propped up by towel rolls at the treatment bedside. The physiotherapist stabilized the talus of the subject with one hand, then placed the other hand’s palm on the medial calcaneus then conducted level III outside sliding. Or the physiotherapist placed the other hand’s palm on the lateral calcaneus and then conduct-ed level III inside sliding.

Each group received 30 seconds of manual therapy, with 1 minute rest between groups. There were 3 groups in total.

Add 4 references:

  1. YU Yue; LIU Dong-sen; RUAN Bing; Qi, G. Advance in Balance Training for Chronic Ankle Instability: A Systematic Review. Chin J Rehabil Theory Pract 2019, 25, 1374-1383.
  2. Luan, L.; Adams, R.; Witchalls, J.; Ganderton, C.; Han, J. Does Strength Training for Chronic Ankle Instability Improve Balance and Patient-Reported Outcomes and by Clinically Detectable Amounts? A Systematic Review and Meta-Analysis. Physical therapy 2021, 101, doi:10.1093/ptj/pzab046.
  3. Shih, Y.F.; Yu, H.T.; Chen, W.Y.; Liao, K.K.; Lin, H.C.; Yang, Y.R. The effect of additional joint mobilization on neuromuscular performance in individuals with functional ankle instability. Physical therapy in sport : official journal of the Association of Chartered Physiotherapists in Sports Medicine 2018, 30, 22-28, doi:10.1016/j.ptsp.2017.12.001.
  4. Cruz-Díaz, D.; Hita-Contreras, F.; Martínez-Amat, A.; Aibar-Almazán, A.; Kim, K.M. Ankle-Joint Self-Mobilization and CrossFit Training in Patients With Chronic Ankle Instability: A Randomized Controlled Trial. Journal of athletic training 2020, 55, 159-168, doi:10.4085/1062-6050-181-18.

8.The groups have a serious imbalance between the number of men and women, why is this? If the study is based on the general population, the number should be equal. This should be named in the "limitations" section.

Thank you for this suggestion.

Three of the subjects withdrew from the study, and the number of men and women was imbalanced, which may affect the statistical efficacy. And the study only used a single-blind design for the participants rather than conducting a double-blind study. Besides, better comparative analysis cannot be achieved due to the lack of a control group with simple manual therapy. Additionally, the hand-held dynamometer used for the muscle strength tests may have introduced human error into the process, skewing the results. So, in future research, objective instruments such as electromyograms and isokinetic dynamometers are supposed to be used in the muscle strength test, to improve experimental rigor.

9.Line 277-289: The results are continuously repeated, changing only the numbers. Please describe them consistently.

Thank you for this suggestion.

In terms of SEBT, after 4 weeks of intervention, there was a statistically significant difference among the three groups: SEBT-A(P<0.001, F=17.453,η2=0.573), SEBT-PM(P=0.001, F=8.584,η2=0.398), SEBT-PL(P=0.006, F=6.324,η2=0.327), SEBT-C(P=0.002, F=7.803,η2=0.375). The post hoc multiple comparisons showed that: there was a statistically significant difference between EG and CG in terms of SEBT-A(P=0.006,95%CI=2.388-14.265) and SEBT-PM(P=0.042,95%CI=0.288-16.712), and there was a statistically significant difference between EG and SG in terms of SEBT-A(P=0.004,95%CI=2.838-14.348), SEBT-PM(P=0.004,95%CI=2.838-14.348), SEBT-PL(P=0.002,95%CI=3.236-14.053), SEBT-C(P=0.003,95%CI=3.229-15.354) (Figure 2A-D).

  1. 1) Line 318. The article does not match the statement. Please add an epidemiology article for this data.

Thank you for this suggestion.

In terms of the improvement of SEBT-A, SEBT-PM, SEBT-PL and SEBT-C, there was a sta-tistically significant difference between EG and SG. This may be due to the Maitland mo-bilization improving the ankle range of movement. After routine rehabilitation, the bal-ance of the ankle was also improved. In the comparison of EG and CG, there was a statis-tically significant difference in terms of SEBT-A and SEBT-PM. Muscle strength training can improve the proprioceptive acuity and balance of the ankle[59].

59.Hanci, E.; Sekir, U.; Gur, H.; Akova, B. Eccentric Training Improves Ankle Evertor and Dorsiflexor Strength and Proprioception in Functionally Unstable Ankles. American journal of physical medicine & rehabilitation 2016, 95, 448-458, doi:10.1097/phm.0000000000000421.

  1. Line 340. Even if the results are favorable, one cannot make such an extreme claim that the intervention has a "more beneficial" impact.

Thank you for this suggestion.

And, the improvement of EG was most obvious, which could be seen that adding Maitland mobilization to routine rehabilitation training for CAI patients may have a more positive effect on balance.

  1. Please include in your conclusions the relationship with the Maitland mobilization.

Thank you for this suggestion.

In summary, the results showed that the balance, ankle range of movement and muscle strength of patients with CAI were improved after three types of intervention methods. And routine rehabilitation training combined with Maitland mobilization can effectively improve the balance ability and ankle range of movement of patients with CAI, but it is not obvious enough to promote the improvement of muscle strength. It is necessary to increase the mid-term and long-term follow-up survey to clarify the effectiveness of Maitland mobilization in CAI in future research.

Reviewer 4 Report

The authors have conducted an interesting study that provides some evidence for the benefit of chronic ankle instability related variables. Given the concerning Maitland mobilization context, such studies are a welcome addition to the literature base. There are some general and specific comments that I’d like the authors to consider.

1. Theoretical and conceptual underpinning. Some theoretical underpinning is required to support the justification for rehabilitation training on chronic ankle instability variables. Further, the concepts of interest require defining consideration of the theoretical underpinning. This is to provide clarity for the reader about what you are measuring and why.

 2. Intervention Research. Given the focus of the study, more consideration of related intervention research is needed within the literature review to help, again, justify the need for the study – by pinpointing the gaps in the existing literature base.

In particular, it is difficult to determine the effect of the improvement because there are a lot of interventions in this paper. Also, I think it's difficult to conclude that this exercise intervention is effective with so few variables.

Author Response

Dear editor,

We deeply appreciate your valuable comments concerning our manuscript. The suggestions are all very helpful for the revision and improvement. We have carefully studied these comments and made related modifications, which we wish could meet with your approval.

  1. Theoretical and conceptual underpinning. Some theoretical underpinning is required to support the justification for rehabilitation training on chronic ankle instability variables. Further, the concepts of interest require defining consideration of the theoretical underpinning. This is to provide clarity for the reader about what you are measuring and why.

Thank you for this suggestion.

SEBT is a standard measurement method for dynamic balance, which also can be conducted to evaluate the dynamic postural control disorder induced by musculoskeletal injuries (such as chronic ankle instability) and assess the risk of injury of lower limbs [34].

WB-DFROM was conducted to evaluate the ankle dorsiflexion motion angle of sub-jects. WB-DFROM was used to simulate the function position of ankle dorsiflexion in daily physical activity[36].

The muscle around the ankle of CAI patients would be wasting and weak, increasing the risk of re-injury[39]. The assessment of muscle strength around the ankle of patients was conducive to the design and improve the rehabilitation plan [40,41].

A sufficient ankle range of movement is necessary for daily physical activity such as walking and running, but the ankle range of movement of patients with CAI was variably limited. Therefore, the primary physical treatment goal was to improve the ankle range of movement [43,44].

CAIT, which consists of 9 questions and involves the self-perception of ankle stability in daily life[49] , was created to evaluate the stability of the ankle. CAIT has shown a good intraclass correlation coefficient (ICC=0.96) [50]with high diagnostic sensitivity and speci-ficity [51].

Add 8 references:

  1. Plisky, P.J.; Rauh, M.J.; Kaminski, T.W.; Underwood, F.B. Star Excursion Balance Test as a predictor of lower extremity injury in high school basketball players. The Journal of orthopaedic and sports physical therapy 2006, 36, 911-919, doi:10.2519/jospt.2006.2244
  2. Zunko, H.; Vauhnik, R. Reliability of the weight-bearing ankle dorsiflexion range of motion measurement using a smartphone goniometer application. PeerJ 2021, 9, e11977, doi:10.7717/peerj.11977.
  3. Keles, S.B.; Sekir, U.; Gur, H.; Akova, B. Eccentric/concentric training of ankle evertor and dorsiflexors in recreational athletes: muscle latency and strength. Scandinavian journal of medicine & science in sports 2014, 24, e29-38, doi:10.1111/sms.12105.
  4. Lin, C.I.; Khajooei, M.; Engel, T.; Nair, A.; Heikkila, M.; Kaplick, H.; Mayer, F. The effect of chronic ankle instability on muscle activations in lower extremities. PloS one 2021, 16, e0247581, doi:10.1371/journal.pone.0247581.
  5. Feger, M.A.; Snell, S.; Handsfield, G.G.; Blemker, S.S.; Wombacher, E.; Fry, R.; Hart, J.M.; Saliba, S.A.; Park, J.S.; Hertel, J. Diminished Foot and Ankle Muscle Volumes in Young Adults With Chronic Ankle Instability. Orthopaedic journal of sports medicine 2016, 4, 2325967116653719, doi:10.1177/2325967116653719.
  6. Rabin, A.; Kozol, Z.; Spitzer, E.; Finestone, A.S. Weight-bearing ankle dorsiflexion range of motion-can side-to-side symmetry be assumed? Journal of athletic training 2015, 50, 30-35, doi:10.4085/1062-6050-49.3.40.
  7. Konor, M.M.; Morton, S.; Eckerson, J.M.; Grindstaff, T.L. Reliability of three measures of ankle dorsiflexion range of motion. International journal of sports physical therapy 2012, 7, 279-287.
  8. Li, Y.; Tsang, R.C.; Liu, D.; Ruan, B.; Yu, Y.; Gao, Q. Applicability of cutoff scores of Chinese Cumberland Ankle Instability Tool and Foot and Ankle Ability Measure as inclusion criteria for study of chronic ankle instability in Chinese individuals. Physical therapy in sport : official journal of the Association of Chartered Physiotherapists in Sports Medicine 2021, 48, 116-120, doi:10.1016/j.ptsp.2020.12.021.
  9. Intervention Research. Given the focus of the study, more consideration of related intervention research is needed within the literature review to help, again, justify the need for the study – by pinpointing the gaps in the existing literature base

Thank you for this suggestion.

The ankle sprain is one of the most common sports injuries and the recurrence rate of the ankle sprain is the highest in lower limbs sports injuries [1,2]. The incidence of ankle sports injuries ranges from 25% to 50 [3,4]. Additionally, 50% of patients with acute ankle sprains did not seek medical assistance, which led to subsequent sprains being repeated and eventually to chronic ankle instability (CAI)[2]. Typical symptoms of CAI include de-creased muscle strength, ligament elasticity, abnormal proprioception, restricted ankle range of movement, etc. Even if they perform routine rehabilitation training at a later stage, 70% of patients would suffer from numerous functional abnormalities[5,6] . CAI severely restricts the exercise ability of the patients so far as to cause traumatic ankle arthritis[7].

Currently, muscle strength training, physical factor treatment, and Kinesio taping combined with external stabilization measures like protective gear, taping, and orthoses, is the primary therapeutic approach for CAI [8-10]. Tsikopoulos K etc. [11]found that ex-ternal stabilization measures did not have more improvement than the controls in dynamic balance through a network meta-analysis. Biz C and Ling W etc. [12,13]found that Kinesio taping did not obviously improve the dynamic balance of CAI patients, especially for ankle proprioception and ankle range of movement, but Kinesio taping can obviously improve the activation of the long peroneal muscle. Luan L etc.[14] explored whether muscle strength training can improve balance performance in patients with CAI. Com-pared with the controls, muscle strength training did not improve the SEBT results. Hall E.A etc.[15] found that the muscle strength of ankle dorsiflexion, varus and eversion was obviously improved after strength training for 6 weeks, and the pain was also relieved.

Therefore, routine rehabilitation training can reduce the degree of swelling and pain after the ankle sprain and enhance muscle strength[16], but it cannot address the possible sequelae of ankle sprain, such as proprioception damage and restricted range of move-ment.

The main characteristic of ankle instability following an ankle sprain, as well as one of the factors contributing to the risk of re-injury, is the restricted range of movement in the ankle, proprioceptive impairment and decreased neuromuscular control ability[17-20]. The ankle range of movement is related to the changes in the subtalar joint and the talocrural joint kinematics, that is, the reduction of talus backward sliding or the change of talus position relative to the ankle point [21]. The disorder of ankle proprioceptive and neuromuscular control ability would induce decreased balance ability in CAI patients [22]. Even after routine rehabilitation training, the preceding symptoms may still be existent.

Manual therapy improves and recovers joint physiological and accessory movements through passive activities. And it can stretch the ligaments and joint capsule around a joint, reduce soft tissue adhesion, realign joints, and recover ligament extensibility [23]. Maitland mobilization was originated by Geoffrey Douglas Maitland, an Australian physiotherapist. It is a crucial diagnostic and therapeutic technique consisting of evalua-tion and treatment for skeletal muscle system dysfunction. The primary purpose of ma-nipulation, which includes rolling, rotating, sliding, and separation traction, is to improve and recover the physiological movement and auxiliary movement of joints [24,25]. In pre-vious studies on musculoskeletal diseases[25-27], manual therapy has been shown to have a positive recovery effect and is regarded as a good way to make the patients com-fortable by increasing the joint range of movement, stimulating nerve conduction path-ways, and improving proprioception [28-31].

To reduce the impact of confounding factors and improve the quality of manual therapy research, researchers have previously conducted comparative studies of real manual therapy and sham manual therapy [32]. The objective of the study was as follows: to add Maitland mobilization to routine rehabilitation training for CAI patients to explore whether there was a more positive improvement than routine rehabilitation alone or rou-tine rehabilitation combined with sham manual therapy, and to determine the effective-ness of Maitland mobilization for CAI patients. We hypothesized that the combination of routine rehabilitation training and Maitland mobilization would have a positive effect on balance, ankle range of movement, and muscle strength in patients with CAI, to explore whether Maitland mobilization is synergistic with routine rehabilitation training to pro-vide a new reference method of treatment for CAI patients.

Add 5 references:

  1. Tsikopoulos, K.; Sidiropoulos, K.; Kitridis, D.; Cain Atc, S.M.; Metaxiotis, D.; Ali, A. Do External Supports Improve Dynamic Balance in Patients with Chronic Ankle Instability? A Network Meta-analysis. Clinical orthopaedics and related research 2020, 478, 359-377, doi:10.1097/corr.0000000000000946.
  2. Biz, C.; Nicoletti, P.; Tomasin, M.; Bragazzi, N.L.; Di Rubbo, G.; Ruggieri, P. Is Kinesio Taping Effective for Sport Performance and Ankle Function of Athletes with Chronic Ankle Instability (CAI)? A Systematic Review and Meta-Analysis. Medicina (Kaunas, Lithuania) 2022, 58, doi:10.3390/medicina58050620.
  3. Ling, W.; Peng, C.; Guanglan, W.; Cheng, Z. Effects of kinesio taping on chronic ankle instability: a systematic review and Meta-analysis. Chinese Journal of Tissue Engineering Research 2023, 27, 2283-2290.
  4. Luan, L.; Adams, R.; Witchalls, J.; Ganderton, C.; Han, J. Does Strength Training for Chronic Ankle Instability Improve Balance and Patient-Reported Outcomes and by Clinically Detectable Amounts? A Systematic Review and Meta-Analysis. Physical therapy 2021, 101, doi:10.1093/ptj/pzab046.
  5. Hall, E.A.; Docherty, C.L.; Simon, J.; Kingma, J.J.; Klossner, J.C. Strength-training protocols to improve deficits in participants with chronic ankle instability: a randomized controlled trial. Journal of athletic training 2015, 50, 36-44, doi:10.4085/1062-6050-49.3.71.

Round 2

Reviewer 2 Report

Dear authors,

the answers to my suggestions are, on the whole, satisfactory. This, together with the indications of the other reviewers, made the article more complete and exhaustive. However, as I said in the first report, I am not convinced by the evaluation methods. I would recommend expanding the limitations section by indicating the future need to confirm the data with instrumental analyzes such as force plates (for balance) and stereophotogrammetry (for range of motion). This specifications, together with the one already reported regarding the evaluation of strength, seem to me to be necessary. Furthermore, in the conclusions section I would suggest the following changes:

1- Line 447: replace "showed" with "suggest"

2- Line 450: replace "can effectively" with "seems to"

Author Response

Dear editor,

We deeply appreciate your valuable comments concerning our manuscript. The suggestions are all very helpful for the revision and improvement. We have carefully studied these comments and made related modifications, which we wish could meet with your approval.

1.I would recommend expanding the limitations section by indicating the future need to confirm the data with instrumental analyzes such as force plates (for balance) and stereophotogrammetry (for range of motion).

Thank you for this suggestion.

Three of the subjects withdrew from the study, and the number of men and women was imbalanced, which may affect the statistical efficacy. And the study only used a single-blind design for the participants rather than conducting a double-blind study. Besides, better comparative analysis cannot be achieved due to the lack of a control group with simple manual therapy. Additionally, the hand-held dynamometer used for the muscle strength tests may have introduced human error into the process, skewing the results. Therefore, in the future research, objective instruments such as electromyograms and isokinetic dynamometers are supposed to be used in the muscle strength test to improve experimental rigor. And it is suggested that force plates and stereophotogrammetry are supposed to be conducted in the tests for balance and joint range of movement.

  1. Line 447: replace "showed" with "suggest"

Thank you for this suggestion.

In summary, the results suggest that the balance, ankle range of movement and muscle strength of patients with CAI were improved after three types of intervention methods.

  1. Line 450: replace "can effectively" with "seems to"

Thank you for this suggestion.

Compared with routine rehabilitation training alone, routine rehabilitation training combined with Maitland mobilization seems to improve the balance ability and ankle range of movement of patients with CAI, but it is not obvious enough to promote the improvement of muscle strength.

Reviewer 3 Report

The authors have resolved all the issues raised.

Please review the following minor revisions:

Introduction:

On lines 40, 42, 45, and 48, cite the article with "et al.", not "etc.".

Materials and methods:

Please do not put the bibliographic citation in the section title or subtitle (line 211, 226, 233). Put the citation in the text.

Results:

Please do not put units in the title or subtitle of the section (line 285, 296, 308, 326).

Put the title of figure 2 correctly (line 306).

Discussion:

Please cite the article with "et al.", not "etc." (line 412).

Reviewer 4 Report

Thank you for your hard work. 

However, there is no answer to item number 2 that I commented on. 

I'll be waiting for your reply.

2. Intervention Research. Given the focus of the study, more consideration of related intervention research is needed within the literature review to help, again, justify the need for the study – by pinpointing the gaps in the existing literature base.

In particular, it is difficult to determine the effect of the improvement because there are a lot of interventions in this paper. Also, I think it's difficult to conclude that this exercise intervention is effective with so few variables.

Author Response

Dear expert,

We are so sorry for the mistake we made, which may have caused a misunderstanding between us. We submitted an answer to the item number 2, but the number of the response was wrong. The following are our latest modifications. Thank you for your tolerance and patience.

1.Intervention Research. Given the focus of the study, more consideration of related intervention research is needed within the literature review to help, again, justify the need for the study – by pinpointing the gaps in the existing literature base

Thank you for this suggestion.

The ankle sprain is one of the most common sports injuries and the recurrence rate of the ankle sprain is the highest in lower limbs sports injuries [1,2]. The incidence of ankle sports injuries ranges from 25% to 50 [3,4]. Additionally, 50% of patients with acute ankle sprains did not seek medical assistance, which led to subsequent sprains being repeated and eventually to chronic ankle instability (CAI)[2]. Typical symptoms of CAI include de-creased muscle strength, ligament elasticity, abnormal proprioception, restricted ankle range of movement, etc. Even if they perform routine rehabilitation training at a later stage, 70% of patients would suffer from numerous functional abnormalities[5,6] . CAI severely restricts the exercise ability of the patients so far as to cause traumatic ankle arthritis[7].

Currently, muscle strength training, physical factor treatment, and Kinesio taping combined with external stabilization measures like protective gear, taping, and orthoses, is the primary therapeutic approach for CAI [8-10]. Tsikopoulos K et al. [11]found that ex-ternal stabilization measures did not have more improvement than the controls in dynamic balance through a network meta-analysis. Biz C and Ling W et al. [12,13]found that Kinesio taping did not obviously improve the dynamic balance of CAI patients, especially for ankle proprioception and ankle range of movement, but Kinesio taping can obviously improve the activation of the long peroneal muscle. Luan L et al.[14] explored whether muscle strength training can improve balance performance in patients with CAI. Com-pared with the controls, muscle strength training did not improve the SEBT results. Hall E.A et al.[15] found that the muscle strength of ankle dorsiflexion, varus and eversion was obviously improved after strength training for 6 weeks, and the pain was also relieved.

Therefore, routine rehabilitation training can reduce the degree of swelling and pain after the ankle sprain and enhance muscle strength[16], but it cannot address the possible sequelae of ankle sprain, such as proprioception damage and restricted range of movement.

The main characteristic of ankle instability following an ankle sprain, as well as one of the factors contributing to the risk of re-injury, is the restricted range of movement in the ankle, proprioceptive impairment and decreased neuromuscular control ability[17-20]. The ankle range of movement is related to the changes in the subtalar joint and the talocrural joint kinematics, that is, the reduction of talus backward sliding or the change of talus position relative to the ankle point [21]. The disorder of ankle proprioceptive and neuromuscular control ability would induce decreased balance ability in CAI patients [22]. Even after routine rehabilitation training, the preceding symptoms may still be existent.

Manual therapy improves and recovers joint physiological and accessory movements through passive activities. And it can stretch the ligaments and joint capsule around a joint, reduce soft tissue adhesion, realign joints, and recover ligament extensibility [23]. Maitland mobilization was originated by Geoffrey Douglas Maitland, an Australian physiotherapist. It is a crucial diagnostic and therapeutic technique consisting of evaluation and treatment for skeletal muscle system dysfunction. The primary purpose of manipulation, which includes rolling, rotating, sliding, and separation traction, is to improve and recover the physiological movement and auxiliary movement of joints [24,25]. In previous studies on musculoskeletal diseases[25-27], manual therapy has been shown to have a positive recovery effect and is regarded as a good way to make the patients comfortable by increasing the joint range of movement, stimulating nerve conduction path-ways, and improving proprioception [28-31].

To reduce the impact of confounding factors and improve the quality of manual therapy research, researchers have previously conducted comparative studies of real manual therapy and sham manual therapy [32]. The objective of the study was as follows: to add Maitland mobilization to routine rehabilitation training for CAI patients to explore whether there was a more positive improvement than routine rehabilitation alone or routine rehabilitation combined with sham manual therapy, and to determine the effective-ness of Maitland mobilization for CAI patients. We hypothesized that the combination of routine rehabilitation training and Maitland mobilization would have a positive effect on balance, ankle range of movement, and muscle strength in patients with CAI, to explore whether Maitland mobilization is synergistic with routine rehabilitation training to provide a new reference method of treatment for CAI patients.

Add 5 references:

  1. Tsikopoulos, K.; Sidiropoulos, K.; Kitridis, D.; Cain Atc, S.M.; Metaxiotis, D.; Ali, A. Do External Supports Improve Dynamic Balance in Patients with Chronic Ankle Instability? A Network Meta-analysis. Clinical orthopaedics and related research 2020, 478, 359-377, doi:10.1097/corr.0000000000000946.
  2. Biz, C.; Nicoletti, P.; Tomasin, M.; Bragazzi, N.L.; Di Rubbo, G.; Ruggieri, P. Is Kinesio Taping Effective for Sport Performance and Ankle Function of Athletes with Chronic Ankle Instability (CAI)? A Systematic Review and Meta-Analysis. Medicina (Kaunas, Lithuania) 2022, 58, doi:10.3390/medicina58050620.
  3. Ling, W.; Peng, C.; Guanglan, W.; Cheng, Z. Effects of kinesio taping on chronic ankle instability: a systematic review and Meta-analysis. Chinese Journal of Tissue Engineering Research 2023, 27, 2283-2290.
  4. Luan, L.; Adams, R.; Witchalls, J.; Ganderton, C.; Han, J. Does Strength Training for Chronic Ankle Instability Improve Balance and Patient-Reported Outcomes and by Clinically Detectable Amounts? A Systematic Review and Meta-Analysis. Physical therapy 2021, 101, doi:10.1093/ptj/pzab046.
  5. Hall, E.A.; Docherty, C.L.; Simon, J.; Kingma, J.J.; Klossner, J.C. Strength-training protocols to improve deficits in participants with chronic ankle instability: a randomized controlled trial. Journal of athletic training 2015, 50, 36-44, doi:10.4085/1062-6050-49.3.71.

2.In particular, it is difficult to determine the effect of the improvement because there are a lot of interventions in this paper. Also, I think it's difficult to conclude that this exercise intervention is effective with so few variables.

Thank you for this suggestion. The objective of the study was as follows: to add Maitland mobilization to routine rehabilitation training for CAI patients to explore whether there was a more positive improvement than routine rehabilitation alone or routine rehabilitation combined with sham manual therapy, and to determine the effectiveness of Maitland mobilization for CAI patients. We hypothesized that the combination of routine rehabilitation training and Maitland mobilization would have a positive effect on balance, ankle range of movement, and muscle strength in patients with CAI, to explore whether Maitland mobilization is synergistic with routine rehabilitation training to provide a new reference method of treatment for CAI patients. Therefore, the main intervention methods in the study are Maitland mobilization and routine rehabilitation training. The sham manual therapy was only used as a control. Routine rehabilitation training is a commonly used intervention at this stage, but it is often overlooked for the improvement of limited joint range of movement. Maitland mobilization is a complete protocol of manual therapy that promotes the improvement of joint range of movement and proprioception. In future research, we will separate the operational steps of Maitland mobilization and examine the operation of each step one by one. Thereby, we will identify which step plays a decisive role and improve the treatment efficiency and effect of Maitland mobilization.